# Why so gloomy? A Bayesian explanation of human pessimism bias in the multi-armed bandit task

**Dalin Guo**
Department of Cognitive Science
University of California San Diego
La Jolla, CA 92093
dag082@ucsd.edu

**Angela J. Yu**
Department of Cognitive Science
University of California San Diego
La Jolla, CA 92093
ajyu@ucsd.edu

## Abstract

How humans make repeated choices among options with imperfectly known reward outcomes is an important problem in psychology and neuroscience. This is often studied using multi-armed bandits, which is also frequently studied in machine learning. We present data from a human stationary bandit experiment, in which we vary the average abundance and variability of reward availability (mean and variance of the reward rate distribution). Surprisingly, we find subjects significantly underestimate prior mean of reward rates – based on their self-report on their reward expectation of non-chosen arms at the end of a game. Previously, human learning in the bandit task was found to be well captured by a Bayesian ideal learning model, the Dynamic Belief Model (DBM), albeit under an incorrect generative assumption of the temporal structure – humans assume reward rates can change over time even though they are truly fixed. We find that the "pessimism bias" in the bandit task is well captured by the prior mean of DBM when fitted to human choices; but it is poorly captured by the prior mean of the Fixed Belief Model (FBM), an alternative Bayesian model that (correctly) assumes reward rates to be constants. This pessimism bias is also incompletely captured by a simple reinforcement learning model (RL) commonly used in neuroscience and psychology, in terms of fitted initial Q-values. While it seems sub-optimal, and thus mysterious, that humans have an underestimated prior reward expectation, our simulations show that an underestimated prior mean helps to maximize long-term gain, if the observer assumes volatility when reward rates are stable, and utilizes a softmax decision policy instead of the optimal one (obtainable by dynamic programming). This raises the intriguing possibility that the brain underestimates reward rates to *compensate* for the incorrect non-stationarity assumption in the generative model and a simplified decision policy.

## 1 Introduction

Humans and animals frequently have to make choices among options with imperfectly known outcomes. This is often studied using the multi-armed bandit task [1, 2, 3], in which the agent repeatedly chooses among bandit arms with fixed but unknown reward probabilities. In a bandit setting, only the outcome of the chosen arm is observed in a given trial. The decision-maker learns how rewarding an arm is by choosing it and observing whether it produces a reward, thus each choice pits exploitation against exploration since it affects not only the immediate reward outcome but also the longer-term information gain. Previously, it has been shown that human learning in the bandit task is well captured by a Bayesian ideal learning model [4], the Dynamic Belief Model (DBM) [5], which assumes the reward rate distribution to undergo occasional and abrupt changes. While DBM assumes non-stationarity, an alternative Bayesian model, the Fixed Belief Model (FBM), assumes the

environmental statistics to be fixed, which is usually consistent with the experimental setting. Previous studies have shown that DBM predicts trial-wise human behavior better than FBM in a variety of behavioral tasks (2-alternative forced choice [5], inhibitory control [6, 7], and visual search [8]), including the bandit task [4] – this occurs despite the task statistics being fixed during the task. While it has been argued that a default assumption of high volatility helps the observer to better adapt to truly volatile natural environments [5], and computationally true change-points are difficult to discern given the noisy binary/categorical observations in these tasks [9], it has nevertheless remained rather mysterious why humans would persist in making this assumption contrary to observed environmental statistics. In this work, we tackle this problem using a revised version of the classical bandit task.

In our experiment, we vary average reward abundance and variability to form four different reward environments. Previous multi-armed bandit studies using binary reward outcomes have typically utilized a neutral reward environment [3, 4, 10, 11, 12], i.e. the mean of the reward rates of all the options across games/blocks is 0.5. Here, we manipulate and partially inform the participants of the true generative prior distribution of reward rates in four different environments: high/low abundance, high/low variance. Notably, the information provided about reward availability is not specific to the arms in a deterministic way as in some previous studies [13], but rather independently affect all arms within the environment. Our goal is to examine how humans adapt their decision-making to different reward environments. Particularly, we focus on whether human participants have veridical prior beliefs about reward rates. To gain greater computational insight into human learning and decision making, we compare several previously proposed models in their ability to capture the human trial-by-trial choices as well as self-report data.

Specifically, we consider two Bayesian learning models, DBM and FBM [5], as well as a simple reinforcement learning model (RL) – the delta rule [14], all coupled with a softmax decision policy. Because FBM (correctly) assumes the reward structure to remain fixed during a game, it updates the posterior mean by weighing new observations with decreasing coefficients, as the variance of the posterior distribution decreases over time. In contrast, by assuming the reward rates to have a (small) fixed probability of being redrawn from a prior distribution on each trial, DBM continuously updates the posterior reward rate distribution by exponentially forgetting past observations, and injecting a fixed prior bias [5, 9]. FBM can be viewed as a special case of DBM, whereby the probability of redrawing from the prior distribution is zero on each trial. RL has been widely used in the neuroscience literature [3, 15, 16], and dopamine has been suggested to encode the prediction error incorporated in the RL model [15, 17]. DBM is related to RL in that the stability parameter of DBM also controls the exponential weights as the learning rate parameter of RL does, but two models are not mathematically equivalent. In particularly, RL has no means of injecting a prior bias on each trial, as DBM does [5, 9]. For the decision policy, we employ a variant of the softmax policy, which is popular in psychology and neuroscience, and has been frequently used to model human behavior in the bandit task [1, 3, 11, 12, 18] – our variant is polynomial rather than the more common exponential form, such that a polynomial exponent of one corresponds to exact "matching" [8], whereas the exponential form has no setting equivalent to matching.

In the following, we first describe the experiment and some model-free data analyses (Sec. 2), then present the model and related analyses (Sec. 3), and finally discuss the implications of our work and potential future directions (Sec. 4).

## 2 Experiment

**Experimental design.** We recruited 107 UCSD students to participate in a four-armed, binary outcomes (success/fail) bandit task, with 200 games in total played by each participant. Each game contains 15 trials, i.e. 15 decision in sequence choosing among the same four options, where reward rates are fixed throughout 15 trials. On each trial, a participant is shown four arms along with their previous choice and reward histories in the game (Fig. 1A); the chosen arm produces a reward or failure based on its (hidden) reward probability. Thirty-two participants were required to report their estimate of the reward rates of the never-chosen arms at the end of each game. Participants received course credits and $0.05 for every point earned in five randomly chosen games (amounts paid ranged from $1.15 to $4.90, with an average of $1.99).

We separated the 200 games into four consecutive environments (sessions), with 50 games each, and provided a clear indication that the participant was entering a new environment. Each environment

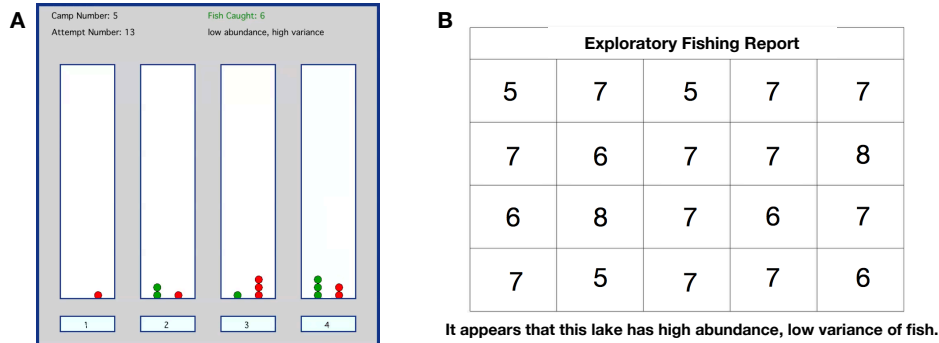

Figure 1: Experimental design. (A) Experimental interface. The total number of attempts so far, total reward, current environment, and the cumulative reward of each option are shown to the subjects on each trial during the experiment. The four panels correspond to the four options (arms). A green circle represents a success (1 point), and a red circle represents a failure (0 point). (B) An example of the fishing report: the numbers represent the total number of fish caught out of 10 attempts at each of the 20 random locations in the environment.

corresponds to a setting of high/low abundance and high/low variance. For any game within a given environment, the reward rates of the four arms are identically and independently pre-sampled from one of four Beta distributions: Beta(4, 2), Beta(2, 4), Beta(30, 15) and Beta(15, 30). These Beta distributions represent the prior distribution of reward rates, where the high/low mean (0.67/0.33) and high/low standard deviation (0.18/0.07) of the distributions correspond to high/low abundance and high/low variability of the environment respectively. The order of the four environments, as well as the order of the pre-sampled reward rates in each environment, are randomized for each subject.

We portrayed the task as an ice fishing contest, where the four arms represent four fishing holes. Participants are informed that the different camps (games) they fish from residing on four different lakes (environments) that vary in (a) overall abundance of fish, and (b) variability of fish abundance across locations. Each environment is presented along with the lake's fishing conditions (high/low abundance, high/low variance) and samples from the reward distribution (a fishing report showing the number of fish caught out of ten attempts at 20 random locations in the lake (Fig. 1B)).

**Results.** The reported reward rates of the never-chosen arms are shown in Fig. 2A. Human subjects reported estimates of reward rate significantly lower than the true generative prior mean ($p < .001$), except in low abundance and low variance environment ($p = 0.2973$). The average reported estimates across the four reward environments are not significantly different ($F(3, 91) = 1.78, p = 0.1570$), nor across different mean, variance or the interaction (mean: $F(1, 91) = 2.93, p = 0.0902$, variance: $F(1, 91) = 0.23, p = 0.6316$, interaction: $F(1, 91) = 1.77, p = 0.1870$). This result indicates that humans do not alter their prior belief about the reward rates even when provided with both explicit (verbal) and implicit (sampled) information about the reward statistics of the current environment. In spite of systematically underestimating expected rewards, our participants appear to perform relatively well in the task. The actual total reward accrued by the subjects are only slightly lower than the optimal algorithm utilizing the correct Bayesian inference and the dynamic-programming-based optimal decision policy (Fig. 2B); humans also perform significantly better than the chance level attained by a random policy ($p < .001$, see Fig. 2B), which is equal to the generative prior mean of the reward rates. Thus, participants experience empirical reward rates higher than the generative prior mean (since they eared more than the random policy); nevertheless, they significantly underestimate the mean reward rates.

## 3   Models

How do humans achieve relatively good performance with an "irrationally" low expectation of reward rates? We attempt to gain some insight into human learning and decision-making processes in bandit task via computational modeling. We consider three learning models, DBM, FBM, and RL, coupled with softmax decision policy. In the following, we first formally describe the models (Sec. 3.1), then compare their ability to explain/predict the data (Sec. 3.2), and finally present simulation results

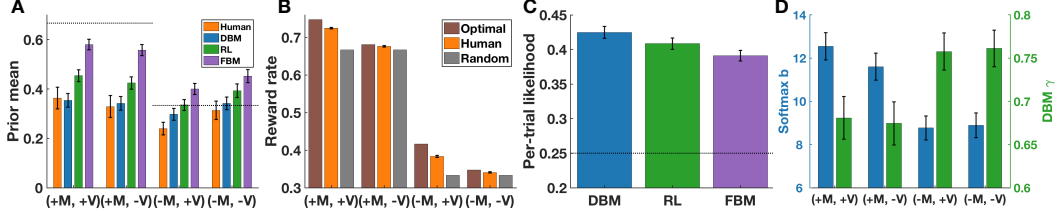

Figure 2: Error bars: s.e.m. across participants or validation runs. (+M, -V) denotes high mean (abundance), low variance, and so on. (A) Reported reward rate estimates by human subjects (orange), and fitted prior mean of DBM (blue), FBM (purple), and RL (green). Dotted lines: the true generative prior mean (0.67/0.33 for high/low abundance environments). (B) Reward rates earned by human subjects (orange), and expected reward rate of the optimal policy (brown) and a random choice policy (gray). (C) Averaged per-trial likelihood of 10-fold cross validation of three learning models. Dotted line: the chance level (0.25). (D) Fitted softmax $b$ and DBM $\gamma$ parameters.

to gain additional insights into what our experimental findings may imply about human cognition (Sec. 3.3, 3.4).

## 3.1   Model description

We denote the reward rate of arm $k$ at time $t$ as $\theta_k^t$, $k \in \{1, 2, 3, 4\}$, $1 \leq t \leq 15$, and $\boldsymbol{\theta}^t = [\theta_1^t, \theta_2^t, \theta_3^t, \theta_4^t]$. We denote the reward outcome at time t as $R_t \in \{0, 1\}$, and $\mathbf{R}^t = [R_1, R_2, \ldots, R_t]$. We denote the decision at time t as $D_t$, $D_t \in \{1, 2, 3, 4\}$, and $\mathbf{D}^t = [D_1, D_2, \ldots, D_t]$.

**Dynamic belief model (DBM).** As with the actual experimental design, DBM assumes that the binary reward (1: reward, 0: no reward) distribution of the chosen arm is Bernoulli. Unlike the actual experimental design, where reward rates of a game are fixed, DBM assumes that the reward rate (Bernoulli rate parameter) for each arm undergoes discrete, un-signaled changes independently with a per-trial probability of $1 - \gamma$, $0 \leq \gamma \leq 1$. The reward rate at time $t$ remains the same with probability $\gamma$, and is re-sampled from the prior distribution $p^0(\theta)$ with probability $1 - \gamma$. The observed data has the distribution, $P(R_t = 1|\boldsymbol{\theta}^t, D_t = k) = \theta_k^t$.

The hidden variable dynamics of DBM is

$$p(\theta_k^t = \theta|\theta_k^{t-1}) = \gamma\delta(\theta_k^{t-1} - \theta) + (1 - \gamma)p^0(\theta), \tag{1}$$

where $\delta(x)$ is the Dirac delta function, and $p^0(\theta)$ is the assumed prior distribution.

The posterior reward rate distribution given the reward outcomes up to time $t$ can be computed iteratively via Bayes' rule as

$$p(\theta_k^t|\mathbf{R}^t, \mathbf{D}^t) \propto p(R_t|\theta_k^t)p(\theta_k^t|\mathbf{R}^{t-1}, \mathbf{D}^{t-1}), \text{ if } D_t = k. \tag{2}$$

Only the posterior distribution of the chosen arm is updated with the new observation, while the posterior distribution of the other arms are the same as the predictive reward rate distribution (see below), i.e. $p(\theta_k^t|\mathbf{R}^t, \mathbf{D}^t) = p(\theta_k^t|\mathbf{R}^{t-1}, \mathbf{D}^{t-1})$, if $D_t \neq k$.

The predictive reward rate distribution at time $t$ given the reward outcomes up to time $t - 1$ is a weighted sum of the posterior probability and the prior probability:

$$p(\theta_k^t = \theta|\mathbf{R}^{t-1}, \mathbf{D}^{t-1}) = \gamma p(\theta_k^{t-1} = \theta|\mathbf{R}^{t-1}, \mathbf{D}^{t-1}) + (1 - \gamma)p^0(\theta), \text{ for all } k. \tag{3}$$

The expected (mean predicted) reward rate of arm $k$ at trial $t$ is $\hat{\theta}_k^t = \mathbb{E}[\theta_k^t|\mathbf{R}^{t-1}, \mathbf{D}^{t-1}]$. DBM can be well approximated by an exponential filter [5], thus $\gamma$ is also related to the length of the integration window as well as the exponential decay rate.

**Fixed belief model (FBM).** FBM assumes that the statistical environment remains fixed throughout the game, e.g. the reward outcomes are Bernoulli samples generated from a fixed rate parameter $\theta$. It can be viewed as a special case of DBM with $\gamma = 1$.

**Reinforcement learning (RL).** The update rule of a simple and commonly used reinforcement learning model is

$$\hat{\theta}_k^t = \hat{\theta}_k^{t-1} + \epsilon(R_t - \hat{\theta}_k^{t-1}), \text{ if } D_t = k. \tag{4}$$

with an initial value $\hat{\theta}_k^0 = \theta^0$, and $0 \le \epsilon \le 1$. $\epsilon$ is the learning parameter that controls the exponential decay of the coefficients associated with the previous observations. In the multi-armed bandit task, only the chosen arm is updated, while the other arms remain the same, i.e. $\hat{\theta}_k^t = \hat{\theta}_k^{t-1}$, if $D_t \ne k$.

**Softmax decision policy.** We use a version of the softmax decision policy that assumes the choice probabilities among the options to be normalized polynomial functions of the estimated expected reward rates [8], with a parameter $b$:

$$p(D_t = k) = \frac{(\hat{\theta}_k^t)^b}{\sum_i^K (\hat{\theta}_i^t)^b}, \tag{5}$$

where $K$ is the total number of options ($K = 4$), $b \ge 0$. When $b = 0$, the choice is at random, i.e. $p(D_t = k) = 1/K$ for all options. When $b = 1$, it is exact matching [8]. When $b \to \infty$, the most rewarding option is always chosen. By varying the $b$ parameter, the softmax policy is able to capture more or less "noisy" choice behavior. However, we note that softmax ascribes unexplained variance in choice behavior entirely to "noise", when subject may indeed employ a much more strategic policy whose learning and decision components are poorly captured by the model(s) under consideration. Thus, smaller fitted $b$ does not imply subjects are necessarily more noisy or care about rewards less; it may simply mean that the model is less good at capturing subjects' internal processes.

**Optimal policy.** The multi-armed bandit problem can be viewed as a Markov decision process, where the state variable is the posterior belief after making each observation. The optimal solution to the problem considered here can be computed numerically via dynamic programming [4, 19], where the optimal learning model is FBM with the correct prior distribution. Previously, it has been shown that human behavior does not follow the optimal policy [4]; nevertheless, it is a useful model to consider in order to assess the performance of human subjects and the various other models in terms of maximal expected total reward.

### 3.2  Model comparison

Here, we compare the three learning models to human behavior, in order to identify the best (of those considered) formal description of the underlying psychological processes.

We first evaluate how well the three learning models fit human data. We perform 10-fold cross-validation to avoid overfitting for comparison, since the models have different numbers of free parameters. We use per-trial likelihood as the evaluation metric, calculated as $\exp(\log \mathcal{L}/N)$, where $\mathcal{L}$ is the maximum likelihood of the data, and $N$ is the total data points. The pre-trial likelihood can also be interpreted as the trial-by-trial predictive accuracy (i.e., on average, how likely is it that the model will choose the same arm the human participant chose), so we will also refer to this measurement as predictive accuracy. We fit prior weight ($\alpha + \beta$, related to precision) at the group level. We fit prior mean ($\alpha/(\alpha + \beta)$), DBM $\gamma$, RL $\epsilon$, and softmax $b$ parameters at the individual level, and separately for the four reward environments. This fitting strategy predicts subjects' choices better than other variants that fit a shared parameter across participants or environments based on 10-fold cross-validation, comparing per-condition fitting and common-parameter fitting. The cross-validation mitigates against the over-fitting issue. Moreover, when the DBM $\gamma$ and softmax $b$ are held fixed across the four conditions for each participant, we find the same pattern of results as when those parameters are allowed to vary across conditions (results not shown).

Fig. 2C shows the held-out per-trial likelihood for DBM, FBM, and RL, averaged across ten runs of cross-validation. DBM achieves significantly higher per-trial likelihood than FBM ($p < .001$) and RL ($p < .001$) based on paired t-test, i.e., predicting human behavior better than the other two models. The predictive accuracy of three fitted models on the whole dataset are 0.4182 (DBM), 0.4038 (RL), and 0.3856 (FBM). DBM also achieves lower BIC and AIC values than RL or FBM, in spite of incurring a penalty for the additional parameter. This result corroborates previous findings [4] that humans assume non-stationarity by default in the multi-armed bandit task, even though the reward structure is truly stationary.

Next, we examine how well the learning models can recover the underestimation effect observed in human participants. The reported estimation is on the arm(s) that they never chose at the end of each game, which is their belief of the mean reward rate before any observation, i.e., mathematically equivalent to the prior mean (DBM & FBM) or the initial value (RL). For simplicity, we will refer to them all as the prior mean. Fig. 2A shows the average fitted prior mean of the models. FBM recovers

Table 1: Inferred RL learning rates

| Model | (+M, +V) | (+M, -V) | (-M, +V) | (-M, -V) |
|---|---|---|---|---|
| RL | 0.35 (SD=0.02) | 0.39 (SD=0.02) | 0.22 (SD=0.02) | 0.25 (SD=0.02) |

Table 2: Inferred softmax b

| Model | (+M, +V) | (+M, -V) | (-M, +V) | (-M, -V) |
|---|---|---|---|---|
| RL | 6.94 (SD=0.54) | 5.41 (SD=0.48) | 5.72 (SD=0.51) | 5.07 (SD=0.51) |
| FBM | 12.28 (SD=0.58) | 10.52 (SD=0.54) | 6.25 (SD=0.42) | 5.68 (SD=0.37) |

prior mean values that are well correlated with the true generative prior means ($r = +.96, p < 0.05$), and significantly different in the four environments ($F(3, 424) = 13.47, p < .001$). The recovered prior means for RL are also significantly different in the four environments ($F(3, 424) = 4.21, p < 0.01$). In contrast, the recovered prior means for DBM are not significantly different in the four environments ($F(3, 424) = 0.91, p = 0.4350$), just like human estimates (Fig. 2A). DBM also recovers prior mean values in low abundance and high variance environment slightly lower than in other environments, similar to human reports. In summary, DBM allows for better recovery of human internal prior beliefs of reward expectation than FBM or RL.

Taking DBM as the best model for learning, we can then examine the other fitted learning and decision-making parameters. A higher softmax $b$ corresponds to a more myopic, less exploratory, and less stochastic choice behavior. A lower DBM $\gamma$ corresponds to a higher change rate and a shorter integration window of the exponential weights [5]. The prior weight of DBM is fitted to six, which is equivalent to six pseudo-observations before the task; it is also the same as the true prior weight in the experimental design for the high variance environments. Fig. 3D shows the fitted DBM $\gamma$ and softmax $b$ in four reward environments. In high abundance environments, softmax $b$ is fitted to larger values, while DBM $\gamma$ is fitted to lower values, than the low abundance environments (four paired t-test, $p < .01$ for all). They do not vary significantly across low- and high-variance environments (four paired t-test, $p > .05$ for all). The fitted DBM $\gamma$ values imply that human participants behave as if they believe the reward rates change on average approximately once every three trials in high-abundance environments, and once every four trials in low-abundance environments (mean change interval is $1/(1-\gamma)$).

### 3.3 Simulation results: shift rate facing an unexpected loss

We simulated the models under the same reward rates as in the human experiment with the parameters fitted to human data, averaged across participants. The fitted DBM $\gamma$ and the fitted softmax $b$ for DBM are shown in Figure. 2D. The fitted learning rate of RL is shown in Table 1, and the fitted softmax $b$ for FBM and RL are shown in Table 2.

To gain some insight into how DBM behaves differently than FBM and RL, and thus what it implies about human psychological processes, we consider the empirical/simulated probability of the participants/model switching away from a "winning" arm after it suddenly produces a loss (Fig. 3A). Since DBM assumes reward rates can change any time, a string of wins followed by a loss indicates a high probability of the arm switching to a low reward rate, especially with a low abundance prior belief, where a switch is likely to result in a new reward rate that is also low. On the other hand, since FBM assumes reward rates to be stable, it depends more on long-term statistics to estimate an arm's reward rate. Give observations of many wins, which leads to a relatively high reward rate estimate as well as a relatively low uncertainty, a single loss should still induce in FBM a high probability of sticking with the same arm. RL can adjust its reward estimate according to unexpected observations, but is much slower than DBM in doing so since it has a constant learning rate that is not increased when there is high posterior probability of a recent change point (as when a string of wins is followed by a loss); RL also cannot persistently encode prior information about overall reward abundance in the environment when a change occurs (e.g. with a low-abundance belief, the new reward rate after a change point is likely to be low). We would thus expect RL to also shift less frequently than DBM in this scenario. Fig. 3A shows that the simulated shift rate of the three models (probability of a model to shift away from the previously chosen arm) exactly follow the pattern of behavior described above.

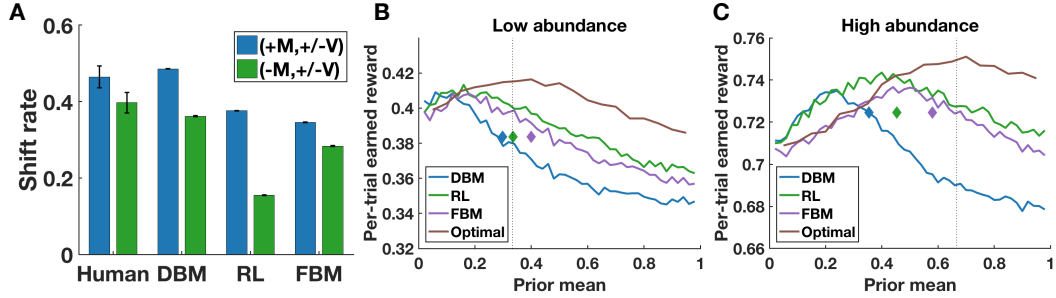

Figure 3: (A) Probability of shifting to other arms after a failure preceded by three consecutive successes on the same arm. Error bar shows the s.e.m. across participants/simulation runs. (B) Reward rates achieved in high variance environment with low abundance and with (C) high abundance by different models: DBM (blue), FBM (purple), RL (green) and optimal policy (brown). The diamond symbols represent the actual reward per trial earned by human subjects (y-axis) vs. the fitted prior mean (x-axis) of the three models. Vertical dotted lines: true generative prior mean.

Human subjects' shift rates are closest to what DBM predicts, which is what we would already expect from the fact that overall DBM has already been found to fit human data the best.

### 3.4  Simulation results: understanding human reward underestimation

Finally, we try to understand *why* humans might exhibit a "pessimism bias" in their reward rate expectation. Fig. 3B,C shows the simulated average earned reward per trial, of the various models as a function of the assumed prior mean. DBM, FBM, RL are simulated with the average parameters that are fitted to human data, except we allow the prior mean parameter to vary in each case. The optimal policy is computed with different prior means and the correct prior variance. The per-trial earned reward rates are calculated from the simulation of models/optimal policy under the same reward rates of the human experiment. We focus on the high variance environments, since the model performance is relatively insensitive to the assumed prior mean in low variance environments (not shown).

Firstly, consider the diamond symbols in Fig. 3B;C: the combination of human subjects' actual average per-trial earned reward (y-axis) and the fitted prior mean for each of the three models (x-axis, color-coded) is very close to DBM's joint predictions of the two quantities (blue lines), but very far away from FBM (purple line) and RL (green line)'s joint predictions of the two quantities. This result provides additional evidence that DBM can predict and capture human performance better than the other two models.

More interestingly, while the optimal policy (brown) achieves the highest earned reward when it assumes the correct prior (as expected), FBM coupled with softmax achieves its maximum reward at a prior mean much lower than the true generative mean. Given that FBM is the correct generative model, this implies that one way to compensate for using the sub-optimal softmax policy, instead of the optimal (dynamic programming-derived) policy, is to somewhat underestimate the prior mean. In addition, DBM achieves maximal earned reward with an assumed prior mean even lower than FBM, implying that even more prior reward rate underestimation is needed to compensate for assuming environmental volatility (when the environment is truly stable). We note that human participants do not assume a prior mean that optimizes the earning of reward (blue diamonds are far from the peak of the blue lines) – this may reflect a compromise between optimizing reward earned and truthfully representing environmental statistics.

## 4  Discussion

Our results show that humans underestimate the expected reward rates (a pessimism bias), and this underestimation is only recoverable by the prior mean of DBM. DBM is also found to be better than FBM or RL at predicting human behavior in terms of trial-by-trial choice and actual rewards earned. Our results provide further evidence that humans underestimate the stability of the environment, i.e. assuming the environment to be non-stationary when the real setting is stationary. This default

non-stationarity belief might be beneficial in real world scenarios in the long run, where the behavioral environment can be volatile [5]. It is worth noting that the best earned per-trial reward rates achievable by DBM and FBM are quite close. In other words, as long as a softmax policy is being used, there is no disadvantage to incorrectly assuming environmental volatility as long as the assumed prior mean parameter is appropriately tuned. Participants' actual assumed prior mean (if DBM is correct) is not at the mode of the simulated performance curve, which underestimates prior mean even more than subjects do. This may reflect a tension to accurately internalize environmental statistics and assume a statistical prior that achieves better outcomes.

Humans often have mis-specified beliefs, even though most theories predict optimal behavior when environmentally statistics are correctly internalized. Humans have been found to overestimate their abilities and underestimate the probabilities of the negative events from the environment [20]. Our result might seem to contradict these earlier findings; however, having a lower prior expectation is not necessarily in conflict with an otherwise optimistic bias. We find that human participants earn relatively high reward rate while reporting low expectation on the unchosen arm(s). It is possible that they are optimistic about their ability to succeed in an overall hostile environment (even though they over-estimate the hostility).

Several previous studies [13, 21, 22] found a human tendency to sometimes under-estimate and sometimes over-estimate environmental reward availability in various behavioral tasks. Major differences in task design complicate direct comparisons among the studies. However, our empirical finding of human reward under-estimation is broadly consistent with the notion that humans do not always veridically represent reward statistics. More importantly, we propose a novel mechanism/principle for why this is the case: a compensatory measure for assuming volatility, which is useful for coping with non-stationary environments and utilizing a cheap decision policy such as softmax. Separately, our work implies that systematic errors may creep into investigators' estimation of subjects' reward expectation when an incorrect learning model is assumed (e.g. assuming subjects believe the reward statistics to be stationary when they actually assume volatility). For future work, it would be interesting to also include an individual measure of trait optimism (e.g. LOT-R), to see if the form of pessimism we observe correlates with optimism scores.

One of the limitations of this study is that human reports might be unreliable, or biased by the experimental design. For example, one might object to our "pessimism bias" interpretation of lower expected reward rates for unchosen alternatives, by attributing it to a confirmation bias [23] or a sampling bias [24]. That is, subjects may report especially low reward estimates for unchosen options, either because they retroactively view discarded options as more undesirable (confirmation bias), or because they failed to choose the unchosen options precisely as they decided these were less valuable to begin with for some reason (regardless of the reason, this scenario introduces a sampling bias of the unchosen arms). However, both of these explanations fail to account for the larger reward under-estimation effect observed in high-abundance environments compared to low-abundance environments, which is also predicted by DBM. Moreover, DBM predicts human choice better than the other models on a trial-by-trial basis, lending us further evidence that the reward under-estimation effect is real.

Another modeling limitation is that we only include softmax as the decision policy. A previous study found that Knowledge Gradient (KG), which approximates the relative action values of exploiting and exploring to deterministically choose the best option on each trial [25], is the best model among several decision policies [4] (not including softmax). However, in a later study, softmax was found to explain human data better than KG [11]. This is not entirely surprising, because even though KG is simpler than the optimal policy, it is still significantly more complicated than softmax. Moreover, even if humans use a model similar to KG, they may not give the optimal weight to the exploratory knowledge gain on each trial, or use a deterministically decision policy. We therefore also conceive a "soft-KG" policy, that adds a weighted knowledge gain to the immediate reward rates in the softmax policy. Based on ten-fold cross-validation, the held-out average per-trial likelihood of softmax and soft-KG, in explaining human data, are not significantly different ($p = 0.0693$), and the likelihoods are on average 0.4248 and 0.4250 respectively. 44 subjects have higher, 44 subjects have lower, and 19 subjects have equal held-out likelihood for soft-KG, compared to softmax. Besides KG, there is a rich literature on the decision-making component [26, 27], as opposed to the learning component that we focus on here. For simplicity, we only included the softmax policy. Future studies with greater statistical power may be able to discriminate between softmax and soft-KG, or other decision policies, but this is beyond the scope and intention of this work.

While this study is primarily focused on modeling human behavior in the bandit task, it may have interesting implications for the study of bandit problems in machine learning as well. For example, our study suggests that human learning and decision making are sub-optimal in various ways: assuming an incorrect prior mean, assuming environmental statistics to be non-stationary when they are stable; however, our analyses also show that these sub-optimalities may potentially be combined to achieve better performance than one might expect, as they somehow compensate for each other. Given that these sub-optimal assumptions have certain computational advantages, e.g., softmax is computationally much simpler than optimal policy, DBM can handle a much broader range of temporal statistics than FBM, understanding how these algorithms fit together in humans may, in the future, yield better algorithms for machine learning applications as well.

### Acknowledgments

We thank Shunan Zhang, Henry Qiu, Alvita Tran, Joseph Schilz, and numerous undergraduate research assistants who helped in the data collection. We thank Samer Sabri for helpful input with the writing. This work was in part funded by an NSF CRCNS grant (BCS-1309346) to AJY.

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
