[Reviews · NeurIPS 2018]

Reviewer 1



This paper presents an intriguing computational dissection of a particular form of reward rate underestimation in a bandit task (what the authors call as "pessimism bias"). Modeling suggests that this bias can be accounted for by a Bayesian model which assumes (erroneously) that reward rates are dynamic. The paper is well-written and the methods are sound. I think it could do a better job relating to previous literature, and there are some questions about the modeling and behavioral analysis which I detail below. Specific comments: I was surprised that there was no mention of Gershman & Niv (2015, Topics in Cognitive Science), which is one of the only papers I'm aware of that manipulates reward abundance. I was also surprised that there was no discussion of relevant work on optimism/pessimism in reinforcement learning, such as Stankevicius et al. (2014, PLOS CB), and in foraging (see for example Bateson 2016, Current Opinion in Behavioral Sciences). Was there a significant interaction between mean and variance for the reported reward rates? It looks like subjects reported significantly higher reward rates in (+M,+V) compared to (-M,+V). The DBM seems to capture the human behavior, but my worry is that by fitting separate parameters for each condition, the model is over-parametrized. Based on the explanation in sections 3.3 and 3.4, shouldn't DBM be able to explain the results without fitting separate parameers for each condition? As the authors point out, it is difficult to interpret softmax temperature parameter estimates because they are at least in part capturing model misfit. So I don't understand why the authors then proceed to interpret these parameter estimates (p. 5). Also, it is well-known that softmax temperature and learning rate trade off against each other, so it is difficult to interpret them separately, yet the authors attempt to do so. There may also be a similar degeneracy between softmax temperature and gamma. Is there any relationship between judged reward rate and individual differences in choice behavior or model parameters? Minor comments: p. 2: "persist making" -> "persist in making" p. 3: "was that is" -> ? Fig 2: It would be helpful to maintain a consistent bar coloring across panels. Fig 2 caption: unnecessary to state that error bars show s.e.m multiple times in the caption. ---------------------- I thought the authors did a good job addressing my comments. My main concern was about the per-condition parameter estimation, but the authors point out in their response that (a) this is supported by cross-validation, and (b) the pattern of results doesn't change when the parameters are held fixed across conditions. I therefore increased my rating by 1 point.

Reviewer 2



I have updated my score in light of the proposed revisions. I like this paper and would be happy if this gets accepted. I liked this paper. The authors show that participants underestimate prior reward rates in a bandit task, a behavior which is well-captured by their dynamic belief model, but not by a fixed belief model and only partially by a traditional RL-model. Interestingly, this initial pessimism might actually be adpative under the assumption that participants expect non-stationary environments and map expectatios onto choices by using a softmax-policy. - I like the experimental paradigm. I would have maybe shown the fishing reports as an icon array with green and red dots just to make it comparable to the actual task. Was there any spatial correlation between participants' mean estimates for different options? It could be that they thought closer options on the lake might produce similar outcomes. - this paradigm also reminds me of a paradigm recently applied by Gershman, in a paper called "Uncertainty and exploration", in which he marked different options as high or low variance to assess participants' exploration strategies - I wonder how this relates to other models of human exploration, for example the finding that humans are often well-captchured by a combination between a Kalman filter learning algorithm and a Thompson sampling exploration strategy (for example, Speekenbrink & Konstantinidis, 2015). Thompson sampling is also known to be equivalent to probability matching, a heuristic that has been frequently called suboptimal but can actually perform quite well under the assumption of nonstationary environment. This also relates back to converging evidence that participants might use a combination of direct and random exploration in multi-armed bandit tasks, and I'm not sure how that can be incoporated by the DBM. I guess a direct incorporation of information gain in the model is similar to a directed exploration strategy, whereas the softmax parameter might track (value-based) random exploration. - would have been nice to ask participants abut their confidence in the estimated reward rate too, just to see if this is more based on uncertainty of the environment or more deliberate - how were participants rewarded in this task? - as a side note, Wu et al. (Generalization and exploration in vast spaces) found that participants tend to under-estimate the spatial correlation of rewards and that this "pessimism bias" turned out to be beneficial, even for algorithms of Bayesian optimization; I think this is quite similar to your temporal finding Minor comments: - I find the coloring scheme for some of the plots a little confusing, sometimes purple is FBM, sometimes it's human data; I would try and make this consistent across plots - I'm not sure how the error bars were calculated, is this a mean over participant-nested estimates? - line 79: p < 10 − 3 should be $10^{-3}$ - Figure 3: I would make all of the plots equi-sized; I'm also not sure how the models were run here, just with the overall mean or with a mean per participant? I guess it's the overall mean, but I would perhaps say that - References: make sure that journal names are capitalized appropriately, for example "computational biology" should be "Computational Biology"

Reviewer 3



Interesting concept but not sure how relatively valuable the contribution is. Quality of the writing is good and creativity of the idea. No issues with novelty and clarity, and the authors have addressed data concerns with comparable experiments within the similar category. Some concerns with the data have already been outlined by the authors. It would have been nice to read further suggestions or solutions to address those concerns.